# Simultaneous presence of *Mycoplasma salivarium* and *Tannerella forsythia* in the implant sulcus after lateral augmentation with autogenous root grafts is associated with increased sulcus probing depth

Karoline Groß[1,2], Didem Sahin[2], Malte Kohns Vasconcelos[1]*, Klaus Pfeffer[1], Frank Schwarz[3], Birgit Henrich[1]

1 Institute of Medical Microbiology and Hospital Hygiene, Heinrich Heine University Düsseldorf, Düsseldorf, Germany, 2 Department of Oral Surgery, University Hospital Düsseldorf, Düsseldorf, Germany, 3 Department of Oral Surgery and Implantology, Carolinum, Johann Wolfgang Goethe University Frankfurt, Frankfurt, Germany

* Malte.KohnsVasconcelos@med.uni-duesseldorf.de

## Abstract

### Objective

To characterize a potential pathogenic role of *Mycoplasma salivarium* and bacterial co-detection patterns on different implant augmentation types.

### Material and methods

36 patients were non-randomly assigned to autogenous lateral alveolar ridge augmentation with either cortical autogenous bone blocks, or healthy autogenous tooth roots or non-preservable teeth. Mucosal inflammation was assessed by probing pocket depth (PD) at all sampling sites and by bleeding on probing (BOP) in a subset of sampling sites, and standardized biofilm samples were obtained from the submucosal peri-implant sulcus and sulcus of a contralateral tooth at two times (t1 after implant placement; t2 after six months). Seven bacterial species were quantified using Taqman PCR.

### Results

Mucosal inflammation did not differ between augmentation groups, but peri-implant sulci showed increased abundance of *M. salivarium* after augmentation with autogenous tooth roots lasting for at least six months (t1 p = 0.05, t2 p = 0.011). In *M. salivarium*-positive samples, *Tannerella forsythia* was correlated with PD (R = 0.25, p = 0.035) This correlation was not observed in *M. salivarium*-negative samples. Compared to all other samples, PD was deeper in co-detection (i.e., simultaneous *M. salivarium* and *T. forsythia*) positive samples (p = 0.022). No association of single or co-detection of bacteria with BOP was observed.

**Data Availability Statement:** All relevant data are within the manuscript and its Supporting Information files.

**Funding:** The study was funded by a grant of the Deutsche Forschungsgemeinschaft (DFG), Bonn, Germany under grant agreement Schwarz 4/2. The titanium implants were provided by Institut Straumann AG, Basel, Switzerland. The funders had no role in study design, data collection and analysis, decision to publish, or preparation of the manuscript.

**Competing interests:** The authors declare that they have no conflict of interests related to this study.

## Conclusion

Presence of *M. salivarium* in peri-implant sulci varies with augmentation method and is associated with increased PD but not BOP. A potential causal role of *M. salivarium* in inflammation through a mechanism involving co-presence of *T. forsythia* requires further study.

## Introduction

Peri-implantitis is a pathological condition occurring in tissues around dental implants, characterized by inflammation in the peri-implant connective tissue and progressive loss of supporting bone [1]. In addition to other signs of inflammation, peri-implantitis sites exhibit increased pocket depths compared to baseline measurements [1]. Diseased implant sites have been shown to be predominantly colonized by gram-negative anaerobic bacteria such as *Prevotella intermedia* and therefore feature microbiological characteristics similar to those noted for chronic periodontal infections [2]. Bacteria with periodontopathogenic potential have been isolated at both healthy and diseased implant sites [3] with a similar distribution, irrespective of the clinical implant status (i.e., healthy, peri- implant mucositis, peri-implantitis). A recent analysis of abundance of 78 species showed increased abundance of 19 bacterial species at peri-implantitis sites compared to healthy implant sites, most prominently *Porphyromonas gingivalis* and *Tannerella forsythia* [4]. Risk factors for the development of periodontitis, a condition closely related to peri-implantitis but affecting natural teeth, include *P. gingivalis* and *Streptococcus mutans* [5]. *P. gingivalis* was shown to be the strongest indicator of generalized aggressive periodontitis and *T. forsythia* was one of the ten best predictors of generalized aggressive periodontitis [6]. The concentration and incidence of *Mycoplasma salivarium* was higher in subgingival biofilm samples and saliva of periodontitis patients than in healthy individuals [7, 8].

Among the mycoplasma species colonising the oral cavity, *M. salivarium* was detected most frequently [9]. *M. salivarium* preferentially resided in dental biofilms and gingival sulci, similar to pathogenic periodontal bacteria [10, 11] and was identified as an opportunistic pathogen in patients with periodontitis [12].

After tooth extraction, jaw bone may degenerate and be lost to a degree that bone augmentation prior to insertion of tooth implants becomes necessary [13–15]. Reconstruction with autogenous bone material is the current standard of care [16]. Alternatively, autogenous tooth root fragments have successfully been used to expand the volume of the alveolar ridge [17, 18]. Changes in microbial colonisation patterns and bacteria detected in peri-implantitis dependent on these implant augmentation methods have not been studied to date.

The study reported on here is a sub-study to an interventional trial assessing the impact of different horizontal ridge augmentation methods on the width of the alveolar ridge [17]. The aim of this study was to assess whether the method of augmentation had an effect on microbiological colonization patterns in general and, more specifically, to characterize presence of *M. salivarium* and bacterial co-detection patterns in different implant augmentation types to understand a potential pathogenic role of *M. salivarium*.

## Materials and methods

### Augmentation methods

As part of the main trial, horizontal ridge augmentation was applied to 38 implants in 36 patients, one patient receiving three implants, in three augmentation groups using either: 1) cortical autogenous bone blocks taken from the retromolar area (standard of care group;

n = 11), 2) healthy autogenous tooth roots like impacted or retained wisdom teeth (n = 14), or 3) not retainable teeth, like endodontically treated or paracortical treated teeth which became loose (n = 13). Treatment group allocation was non-random, i.e., was based on availability of root material and investigator judgement. Preparation of tooth-derived augmentation materials was conducted as described previously [17].

Implants were placed 26 weeks after insertion of the horizontal ridge augmentation. During this interval, healing of the operational area was either open or submerged. In case of submerged healing the re-entry was performed 13 weeks after implant placement. If the healing was open, the re-entry was after 10 weeks. Afterwards the prosthetic treatment began with abutments/gingiva former.

## Study population

The patients studied were originally included into an interventional trial to which a detailed description has been published previously [17]. All trial participants also participated in this sub-study. Detailed eligibility criteria for the trial have been described previously [17]. In short, patients meeting the following conditions were included in the trial: (a) age between 18 and 60 years, (b) lateral ridge augmentation necessary based on judgement of the clinician, (c) insufficient bone ridge width at the recipient site for implant placement based on cone beam computed tomography, (d) sufficient bone height at the recipient site for implant placement, (e) healthy oral mucosa and (f) possibility of extraction of retainable tooth/teeth (for groups receiving autogenous tooth roots only).

Patients were not included in the trial if they presented with any one of the following conditions: (a) general contraindications for dental and/or surgical treatments, (b) inflammatory and autoimmune disease of the oral cavity, (c) uncontrolled diabetes (HbA1c > 7%), (d) history of malignancy requiring chemotherapy or radiotherapy within the past five years, (e) previous immuno-suppressant, bisphosphonate or high dose corticosteroid therapy, (f) smokers and (g) pregnant or lactating women.

Sample size calculation, as explained in detail in [19], was performed for the main outcome of the intervention trial and did not include considerations for the study reported here.

## Biofilm samples

At trial entry, all patients received a supra-mucosal cleaning. All samples were collected by the same staff member of the Department of Oral Surgery, Heinrich Heine University, according to a standard operating procedure. Submucosal peri-implant sulcus fluid samples were taken at the deepest aspect of each implant site with sterile paper points (ISO 35–40) (VDW, Munich, Germany) left in place for 30s (implant). The same method was applied for the collection of submucosal biofilm samples at a contralateral tooth (tooth). Samples were taken 36 weeks after implantation (i.e., Time point (t1) at beginning of the prosthetic treatment) and 6 months later (t2), both taken in the same gingival pockets. In addition, a parodontometer was used to detect the pocket depth where samples were taken. Bleeding on manual probing (BOP) was documented at the deepest aspect of each implant site, but not on tooth sites. The paper points were transferred into 200μl G2 buffer solution from the EZ1 DNA Tissue Kit (Qiagen, Hilden, Germany) and stored between -20˚C and -80˚C until transportation to the microbiological laboratory for analysis.

## Genomic DNA preparation

Material attached to the paper points was resuspended by vortexing in the 200μl G2 storage buffer after addition of further 60μl G2 buffer. 200μl of the suspension was supplemented with

12,5μl Proteinase K solution (100μg/ml Proteinase K) and incubated for 30 min at 56˚C. Proteinase K was then inactivated for 5 min at 95˚C. The specimens were homogenized and cells mechanically disrupted by bead beating with 1.4mm ceramic beads (Peclab; Pecelly lysing Kit K14_0,5ml, VWR International GmbH, Darmstadt, Germany) for 4 min at 5.000rpm by using the Minilys Personal Homogenizer (Bertin GMBH, Frankfurt am Main, Germany) [20]. Total genomic DNA isolation was performed by a semiautomatic DNA preparation using an EZ1 biorobot machine (Qiagen, Hilden, Germany) with an elution volume of 50μl. The eluate was stored at -20˚C until further use.

## TaqMan Polymerase Chain Reaction (PCR)

In house TaqMan PCRs for the quantification of *Mycoplasma salivarium* [21], *Veillonella parvula* [22], *Staphylococcus aureus* [23], *Porphyromonas gingivalis* [24], *Parvimonas micra* [25], *Tannerella forsythia* [24], and total eubacterial load (Eubacteria-PCR (Euba)) (S1 Table) were carried out in a total volume of 25μl consisting of 1x Eurogentec qPCR MasterMix (Eurogentec, Seraing Belgium) without ROX (containing buffer, dNTPs (including dUTP), HotGOldStar DNA polymerase, 5 mM $MgCl_2$, uracil-N-glycosylase and stabilizers), 300 nM each forward and reverse primer, 200 nM labelled probe, and 2,5 μl of template DNA (primer and probes are listed in S1 Table) [26]. Amplicon carrying plasmids were used as quantification standards in concentrations of $10^5$, $10^3$ and $10^2$ copies/μl for bacterial species detections and $10^7$, $10^5$ and $10^4$ copies/μl for total bacterial load. Thermal cycling conditions were as follows: 1 cycle at 95˚C for 10min followed by 45 cycles at 95˚C for 15s, and 60˚C for 1min. According to the manufacturer's instructions, cycling and fluorescence measurement and analysis were carried out with an iCycler from BioRad (Bio-Rad CFX Manager 3.1; Bio-Rad, Hercules, CA,).

## Statistical analysis

Statistical analysis was performed using GraphPad Prism Version 5.01 (GraphPad Software Inc., CA). The study reported here was planned as an exploratory addition to the interventional trial reported previously [19] and as such, the analysis design was exploratory without pre-specified hypotheses or a pre-study power calculation.

Bacterial loads were log transformed for analysis and graphical display. Species of bacteria were coded as present if they were detected on qPCR regardless of their abundance and as absent if they were not detected. A gingival pocket depth of >3mm was regarded as indicative of disturbed milieu.

The nonparametric Kruskal-Wallis-Test was used to assess differences in bacterial loads across augmentation groups, and the nonparametric Mann-Whitney for any comparisons between two groups. Categorical variables as above were compared over treatment groups by Chi square tests. Spearman correlation was used to assess the strength of association between continuous variables, i.e., bacterial quantities and pocket depth. To assess the independent association of bacterial species with pocket depth, we fitted (1) a (fixed-effects) linear regression model of *M. salivarium* bacterial load on pocket depth with *T. forsythia* as a co-variate and (2) to include in this model interfering random effects from tooth or implant side, treatment group and timepoint of sampling, a four-level mixed-effects model with random effects and independent covariance in the aforementioned order. The fixed- and mixed-effects models were calculated in Stata Release 14 (StataCorp, College Station, TX).

## Ethical approval

The study protocol was approved by the ethics Committee of Heinrich Heine University Düsseldorf (Ethics Approval Number 6247R) and all patients gave written informed consent.

## Results

### Differences between augmentation groups

Total bacterial load did not differ between augmentation groups (S1 Fig). When comparing quantities of established paropathogens tested between the treatment groups, there were no differences between the augmentation groups for *T. forsythia* (Fig 1B), *Parvimonas micra*, *Staphylococcus aureus* and *Veillonella parvula* (S1 Fig).

*P. gingivalis* was only detected in patients of treatment groups 2 and 3, where new augmentation techniques were used (Fig 1C). On the tooth side, this finding is supported by some statistical evidence (t1 p = 0.0322); t2 (p = 0.0146), whereas on the implant side the number of positive samples was too low for the difference to be statistically significant.

There were specific differences in quantities of *M. salivarium* between the augmentation groups: at t1, the amount of *M. salivarium* differed between augmentation groups and was highest in group 2 where healthy autogenous tooth roots were used for augmentation (Fig 1A). The observed pattern was similar on tooth and implant side with good statistical evidence for a difference between groups on the tooth side (p = 0.01) and weak evidence on the implant side (p = 0.05).

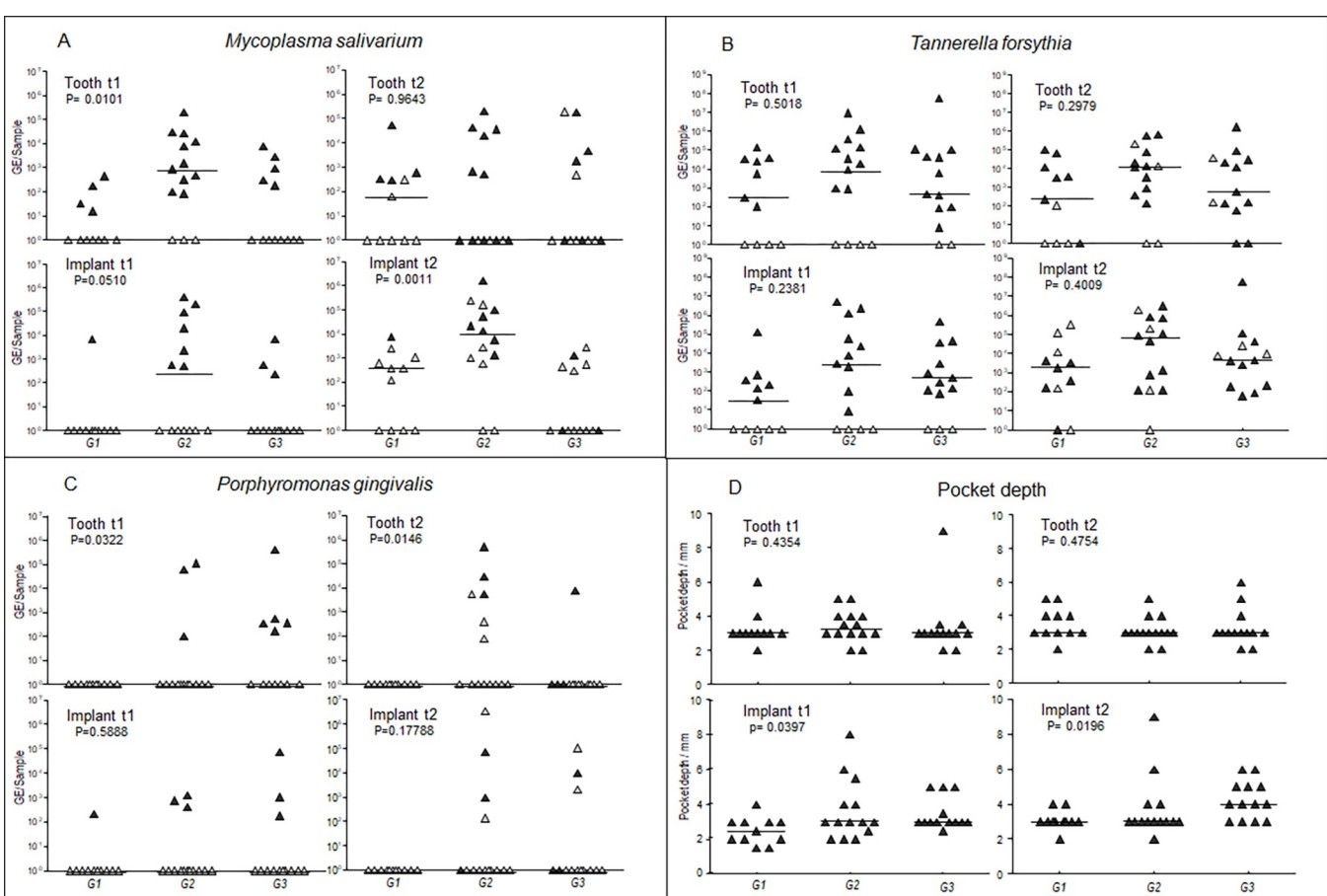

**Fig 1. Abundance of bacterial species and gingival pocket depth by underlying augmentation material.** Scatter plots of the bacterial quantity of *(A) Mycoplasma salivarium, (B) Tannerella forsythia* and (C) *Porphyromonas gingivalis;* indicated as genome equivalents per sample (GE/sample) and of the (D) pocket depth /mm in the peri-implantary sulcus compared to the opposite tooth, over time in the different augmentation groups. G1: Group 1, cortical autogenous bone blocks; G2: Group 2, healthy autogenous tooth roots; G3: Group 3, roots from non-preservable tooth; t1: begin of the prosthetic restauration; t2: six months after completing of the prosthetic restauration. ▲ = positive patient sample at time t1; △ = negative patient sample at time t1. The bars represent mean.

At t2, the difference between treatment groups remained constant on the implant side (p = 0.011) but was no longer seen on the tooth side (p = 0.964) (Fig 1A). The differences between treatment groups followed the same pattern when, instead of quantity, presence or absence of *M. salivarium* was compared between groups (S2 Table).

### Changes in *M. salivarium* abundance over time

Changes in *M. salivarium* abundance between time points t1 and t2 followed different patterns between augmentation groups and between tooth and implant side. In detail, the following observations were made:

On the tooth side, in patients of the standard of care group (group 1), the number of patients with detectable amounts of *M. salivarium* increased between t1 and t2, and all patients who were already positive at t1 were also positive at t2 (Fig 1A). In group 2, augmented with impacted or retained wisdom tooth roots, there was a decrease in the number of positive patients between t1 (n = 11) and t2 (n = 6). In group 3, augmented with not retainable tooth roots, the number of samples positive for *M. salivarium* remained the same between t1 and t2, but two patients who were negative at t2 became positive and vice versa.

On the implant side, the number of positive *M. salivarium* patients in all groups increased between t1 and t2 (Fig 1A). There was no marked difference in the quantity of *M. salivarium* in any of the three groups between t1 and t2.

To exclude that the differences in abundance of *M. salivarium* were due to healing procedure, we compared healing procedures by augmentation groups. Open healing was not used in the standard of care group (group1) and only in one case in study group 2 but was used in almost half of cases in study group 3. The use of healing procedure was therefore distributed differently from the abundance of *M. salivarium*.

### Correlation of gingival pocket depth with different pathogens

To further characterize a possible role of *M. salivarium* in gingival inflammation, we investigated associations between pocket depth and abundance of *M. salivarium*, as well as known paropathogens *P. gingivalis* and *T. forsythia*.

Pocket depth did not differ between groups at the different points in time or between tooth and implant side, and was mostly in a healthy range with a median depth of 3mm. Only at t2 on the implant side in the group augmented with not retainable root grafts (group 3), the median depth was 4mm. (Fig 1D).

No correlation between pocket depth and amount of *P. gingivalis* was observed (R = 0.05, p = 0.530). (Fig 2B and 2C). As expected for a known paropathogen, there was a weak correlation between pocket depth and amount of *T. forsythia* (R = 0.22, p = 0.006). Pocket depth was also weakly correlated with the bacterial load of *M. salivarium* (R = 0.18, p = 0.0305), meaning that a higher quantity of *M. salivarium* tended to be found in deeper pockets (Fig 2A). The amounts of *M. salivarium* and *T. forsythia* were borderline weakly-to-moderately correlated (R = 0.39, p<0.001) (Fig 2D). Due to the correlation of amount of *T. forsythia* both with pocket depth and with amount of *M. salivarium*, it was necessary to investigate whether the correlation between amount of *M. salivarium* and pocket depth was independently observed. Therefore, the association analysis between abundance of *M. salivarium* and pocket depth was repeated adjusting for different concomitant amounts of *T. forsythia*, meaning that an independent association should only be stated if abundance of *M. salivarium* correlated with pocket depth in concordant direction in the co-presence of different amounts of *T. forsythia*. Covariate linear regression showed that after adjusting for amount of *T. forsythia* there was no evidence for an independent association of *M. salivarium* and pocket depth (p = 0.375).

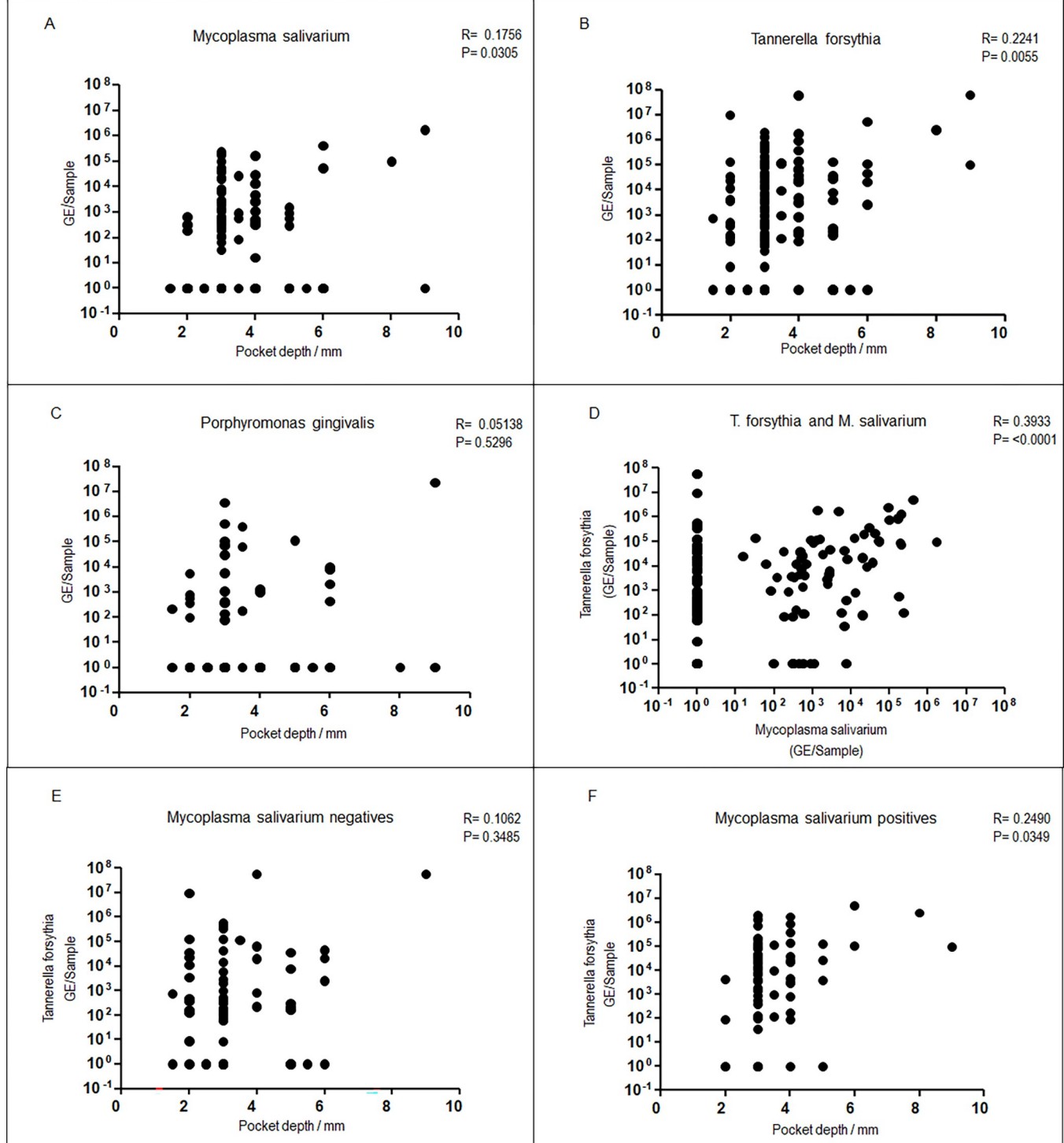

**Fig 2. Correlation of bacterial abundance in the peri-implant sulcus with pocket depth or between bacterial species.** Correlation of the load (GE/sample) of *M. salivarium* (A), *T. forsythia* (B) and *P. gingivalis* (C) to pocket depth (mm); of *T. forsythia* to *M. salivarium* (D); of *M. salivarium* negative samples (E) *or M. salivarium* positive samples (F) to pocket depth (mm). The bars represent mean.

Equally, allowing for random effects on the levels of side of sampling (tooth or implant), treatment group and timepoint of sampling (nested in that order) showed a maintained association between *T. forsythia* and pocket depth (p = 0.017) and no independent association between *M. salivarium* and pocket depth (p = 0.336).

## Modification of the association between abundance of *T. forsythia* and gingival pocket depth by presence or absence of *M. salivarium*

We observed a marked difference in the association of pocket depth and the amount of *T. forsythia* depending on the presence or absence of *M. salivarium*. In the absence of *M. salivarium* (80 samples) there was no association between the amount of *T. forsythia* and pocket depth (R = 0.11, p = 0.349), while in the presence of *M. salivarium* (72 samples) there was a weak association (R = 0.25, p = 0.035) (Fig 2E and 2F).

Next, pocket depth was compared between groups with different co-detection patterns, depending on the presence or absence of *M. salivarium* and *T. forsythia* in the peri-implant sulcus and the contralateral tooth, i.e., detection of either both species simultaneously (co-detection), each one in the absence of the other or of neither species. Overall, no evidence for a non-random distribution of pocket depth between groups could be found. When grouped into co-detection positive or negative (this group encompassing all samples either negative for both or positive only for either *M. salivarium* or *T. forsythia*), gingival pockets were deeper in co-detection positive samples (p = 0.022). Yet, between *T. forsythia* positive/*M. salivarium* negative and co-detection positive the difference was less pronounced (p = 0.070) (Fig 3A). Between *T. forsythia* positive/*M. salivarium* negative and co-detection positive samples, the proportion of deeper gingival pockets indicating inflammation was higher in the latter (30.9% vs. 38.1%) but given the small sample size we cannot exclude that this difference occurred randomly. When comparing the distribution of samples with *M. salivarium* and *T. forsythia* co-detection between treatment groups, no consistent effect of augmentation group was observed (Table 1).

## Bleeding on probing and how it corresponds to the previous observations

There was a weak evidence for deeper pockets in BOP-positive peri-implant locations (p = 0.060), which can characterise a peri-implant mucositis by mucosal inflammation in

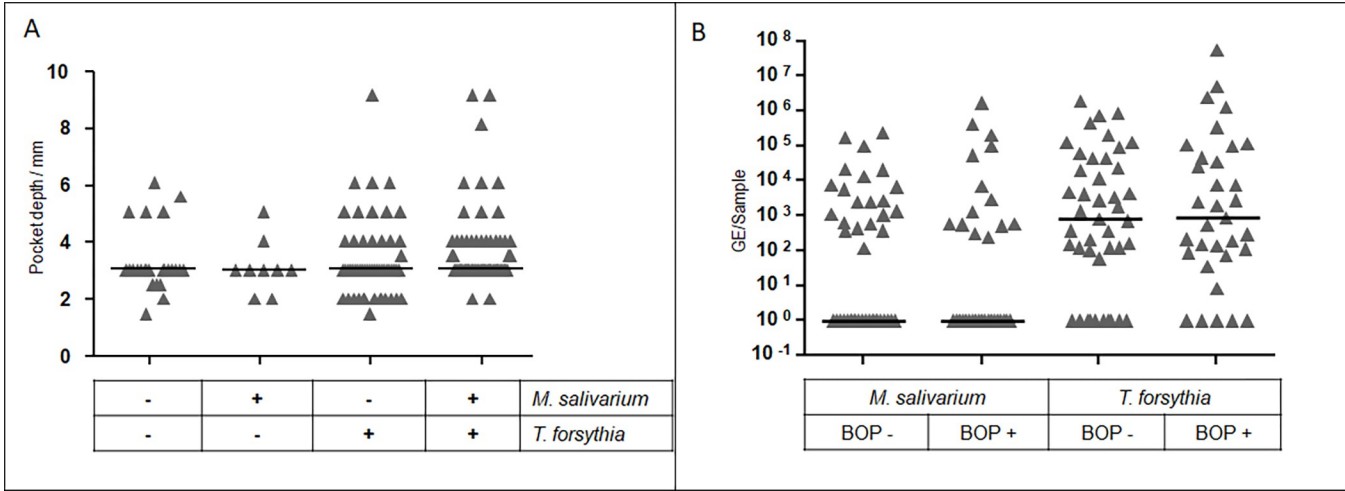

**Fig 3. Association between *T. forsythia* and *M. salivarium* and clinical signs of gingival inflammations.** Scatter plot of the (A) pocket depth (mm) and (B) bleeding on probing positive (BOP+) or negative (BOP-) samples divided in groups of presence (+) and/or absence (-) of *M. salivarium* and *T. forsythia*. The bars represent mean.

**Table 1. Numbers and proportions of simultaneously *Mycoplasma salivarium* and *Tannerella forsythia* positive submucosal biofilm and peri-implant sulcus fluid samples.**

| | Group 1/CABB (n = 11) | Group 2/HATR (n = 14) | Group 3/NPTR (n = 13) | P |
|---|---|---|---|---|
| Tooth side t1 | 3 (27.3%) | 8 (57.1%) | 3 (23.1%) | 0,137 |
| Tooth side t2 | 5 (45.5%) | 6 (42,9%) | 5 (38.5%) | 0,940 |
| Implant side t1 | 1 (9.1%) | 6 (42.9%) | 3 (23.1%) | 0,155 |
| Implant side t2 | 6 (54.6%) | 12 (85.7%) | 5 (38.5%) | 0,038 |

*P*-values obtained by Chi square test; Group 1: cortical autogenous bone blocks (CABB); Group 2: healthy autogenous tooth roots (HATR); Group 3: roots from non-preservable teeth (NPTR); t1: beginning of the prosthetic restauration, t2: six months after completing of the prosthetic restauration.

absence of continuous marginal peri-implant bone loss. Mucositis is considered a precursor for peri-implantitis.

To confirm if *M. salivarium* or *T. forsythia* were associated with peri- implant mucositis, we looked at the association between BOP and bacterial load. There is no evidence for an association between BOP and the bacterial abundance or presence of *M. salivarium* or *T. forsythia* or the co-detection of both bacteria (Fig 3B).

## Discussion

In this study, the underlying augmentation material had no effect on the total abundance of bacteria present in the implant sulcus after lateral alveolar ridge augmentation, but did alter the abundance of some bacterial species tested. For *M. salivarium* specifically, augmentation with autogenous tooth roots led to increased abundance lasting for at least six months. Peri-implant gingival pocket depth correlated with abundance of *T. forsythia* independently of simultaneous amounts of *M. salivarium*, while abundance of *M. salivarium* was not independently correlated with pocket depth when accounting for simultaneous amounts of *T. forsythia*. On more detailed examination, abundance of *T. forsythia* correlated with pocket depth only in the presence of *M. salivarium*, but not in its absence. *T. forsythia* has previously been established as a paropathogen and may therefore independently contribute to development of deeper probing depth, while *M. salivarium* may (a) either be an indicator of increased pathogenicity of *T. forsythia*, (b) may increase the pathogenicity of *T. forsythia* or (c) may itself express pathogenic potential only in the presence of *T. forsythia*.

*M. salivarium* has previously been described as an oral colonizer, localized in the epithelial cells of oral leucoplakia tissue [27, 28], lichen planus [29], in a submasseteric abscess [30], and on the surface of squamous cell carcinoma [28]. The presence of *M. salivarium* has been reported in root canals of patients who needed endodontic treatment [31]. The study reported here demonstrates that after augmentation with healthy tooth roots abundance of *M. salivarium* was persistently increased in the sulcus of root graft augmented implants. Wide anatomical variation of wisdom teeth, which among others [32] were used in this study in group 2, could make preparation of the root grafts augmentation during lateral augmentation more difficult, and thus increase tissue residuals in the augmentation material that may facilitate bacterial colonization. *M. salivarium* has repeatedly been shown to be more frequently present in inflamed gingival sulci and to be present in higher amounts in inflamed sulci than in healthy, non-inflamed sulci [9–11, 33]. A correlation between pocket depth and amount of *M. salivarium* in patients with chronic periodontitis disease has been demonstrated [34], but simultaneous presence of *T. forsythia* was not tested for in the respective study. Based on our results, it is likely that *T. forsythia* may have contributed to periodontitis in the patients described there.

In a C57BL/6 (B6) mouse model, induction of IL-1β in macrophages and dendritic cells after priming of these cells with lipopolysaccharids (LPS) derived from Gram-negative bacteria has been shown and suggested as a possible mechanism for co-pathogenicity of *M. salivarium* and Gram negative paropathogens [33]). *T. forsythia*, a Gram-negative bacterium, was frequently detected in deep periodontal pockets co-localizing with *P. gingivalis* and *T. denticola* to the superficial layers of subgingival biofilm as microcolony blooms adjacent to the pocket epithelium, suggesting possible inter-bacterial interactions that contribute towards disease [35].

The present study has some important limitations. We detected *M. salivarium* and *T. forsythia* at increased abundance in deeper pockets, but their presence or abundance was not associated with BOP. While pocket depth ≥4mm is a strong and commonly used marker of periodontitis, previous studies showed no significant difference in BOP positivity between peri-implant and contra-lateral dental sites when controlling for the difference in PD [36]. Bleeding on probing was agreed as the primary measure of acute inflammation [37, 38]. Deeper pockets did not indicate acute mucositis, which is characterised as BOP on at least one aspect of the implant but without concomitant increases in PD compared to baseline [39]. A peri-implant mucositis is considered a precursor for peri-implantitis [39]. In BOP-positive peri-implant locations weak evidence for deeper pockets could be found, which is also weak evidence for peri-implantitis, characterized by inflammation in the peri-implant connective tissue and progressive loss of supporting bone [1]. In our study, there was no evidence for an association between BOP and the bacterial abundance or presence of *M. salivarium* or *T. forsythia* or co-detection of both.

Any possible causal contribution of *M. salivarium* and *T. forsythia* acute gingival inflammation cannot be derived from the presented data. An inflammation can, however, precede the deeper pockets, and thus create a good environment for the bacteria in deeper pockets through bone breakdown. It remains to be investigated whether the BOP of the implants compared to the tooth side behaves differently due to the underlying augmentation material.

Further, assignment of augmentation method was not random, therefore uncontrolled confounding may have influenced the results. However, clinical characteristics of the augmentation groups in this study have been presented before [40], and clinically important differences between the augmentation groups were not seen. Therefore, although randomization would have been preferable, we believe that the augmentation groups were clinically similar enough to allow the comparisons made in this report. Further, the sample size of the study was primarily chosen for clinical comparisons and although statistical power in this study was high enough to generate good statistical evidence for the microbiological findings reported here, additional and less pronounced microbiological differences between the augmentation groups may have been missed due to the small sample size.

The results presented here on co-localization of *M. salivarium* and *T. forsythia*, including the evidence for an interaction between these species, and the subsequent implications on probing depth are observational. However, for an experimental design *in vivo*, it would be necessary to specifically alter presence and abundance of one or the other species. Specific human infections studies would be ethically unacceptable and specific depletion of one or the other species is unfeasible. We therefore conclude that an observational approach, as the one described here, is the only feasible way to study these interactions *in vivo* in humans. The findings we describe are consistent across the different comparisons and statistical tests presented in this report and fit in with previous studies. A subsequent experimental study, e.g., in an animal model, should measure BOP in all sampling locations and ideally involve radiographic assessment of implant status and could thereby confirm our findings and present a clearer understanding of the mechanisms involved.

Clinical success of lateral root graft augmentation has already been demonstrated in previous studies [17, 18] and a previous report from this study confirms that the implants were healthily retained until the end of follow-up, i.e. 37 to 54 weeks after Implantation [40]. However, none of these reports described microbiological colonization at implantation sites.

The present study is, to the best of our knowledge, the first to demonstrate differences in microbial colonization dependent on implant augmentation methods. The reported association between co-detection of *M. salivarium* and *T. forsythia* in deeper pockets suggests that *M. salivarium* may be of importance in the development of peri-implantitis in addition to its previously suggested role in periodontitis. Future studies on implant augmentation methods should include microbiological parameters and could be extended to use of metagenomic sequencing techniques to gain more comprehensive insights into the peri-implant microbiome. Especially, further mechanistic research into co-pathogenesis between *M. salivarium* and *T. forsythia* is warranted. Understanding of these patho-physiological processes will have the potential to improve long-term success of implantation methods by providing a basis for adjunct prophylactic or therapeutic interventions.

## Supporting information

**S1 Table. Overview of bacterial species, corresponding genes, primers/probes, and DNA sequences as published [26].**
(DOCX)

**S2 Table. Number of *Mycoplasma salivarium* positive submucosal biofilm and peri-implant sulcus fluid samples by underlying augmentation material by intervention group.** Group 1: cortical autogenous bone blocks (CABB); Group 2: healthy autogenous tooth roots (HATR); Group 3: roots from non-preservable teeth (NPTR); t1: beginning of the prosthetic restauration, t2: six months after completing of the prosthetic restauration.
(DOCX)

**S1 Fig. Total bacterial load by underlying augmentation material.** Scatter plots of the bacterial quantity of (A) the Bacterial load, (B) *Parviromonas micra*, (C) *Veillonella parvula and (D) Staphylococcus aureus* indicated as genome equivalents per sample (GE/sample) in the peri-implantary sulcus compared to the opposite tooth, over time in the different augmentation groups. G1: Group 1, cortical autogenous bone blocks; G2: Group 2, healthy autogenous tooth roots; G3: Group 3, roots from non-preservable tooth; t1: begin of the prosthetic restauration; t2: six months after completing of the prosthetic restauration The bars represent mean.
(TIF)

**S1 Dataset. Minimal anonymised dataset.**
(XLSX)

## Acknowledgments

The authors thank Dana Belick, Meike Rosenblatt and Sebastian Scharf (Institute of Medical Microbiology and Hospital Hygiene, Heinrich Heine University, Düsseldorf) for support and assistance in DNA preparation and PCR.

## Author Contributions

**Conceptualization:** Frank Schwarz, Birgit Henrich.

**Data curation:** Karoline Groß.

**Formal analysis:** Karoline Groß, Malte Kohns Vasconcelos.

**Investigation:** Karoline Groß, Didem Sahin, Frank Schwarz, Birgit Henrich.

**Methodology:** Karoline Groß, Malte Kohns Vasconcelos, Frank Schwarz, Birgit Henrich.

**Project administration:** Frank Schwarz, Birgit Henrich.

**Resources:** Klaus Pfeffer, Frank Schwarz, Birgit Henrich.

**Supervision:** Klaus Pfeffer, Frank Schwarz, Birgit Henrich.

**Validation:** Malte Kohns Vasconcelos.

**Visualization:** Karoline Groß, Malte Kohns Vasconcelos.

**Writing – original draft:** Karoline Groß, Malte Kohns Vasconcelos, Birgit Henrich.

**Writing – review & editing:** Didem Sahin, Klaus Pfeffer, Frank Schwarz.

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
