## [Decision Letter · Decision Letter 0]

26 Oct 2021

PONE-D-21-31526Simultaneous presence of Mycoplasma salivarium and Tannerella forsythia in the sulcus of lateral autogenous root grafts augmented implants is associated with increased sulcus probing depthPLOS ONE

Dear Dr. Vasconcelos,

Thank you for submitting your manuscript to PLOS ONE. After careful consideration, we feel that it has merit but does not fully meet PLOS ONE’s publication criteria as it currently stands. Therefore, we invite you to submit a revised version of the manuscript that addresses the points raised during the review process. Please submit your revised manuscript by Dec 10 2021 11:59PM. If you will need more time than this to complete your revisions, please reply to this message or contact the journal office at plosone@plos.org. Please include the following items when submitting your revised manuscript:A rebuttal letter that responds to each point raised by the academic editor and reviewer(s). You should upload this letter as a separate file labeled 'Response to Reviewers'.A marked-up copy of your manuscript that highlights changes made to the original version. You should upload this as a separate file labeled 'Revised Manuscript with Track Changes'.An unmarked version of your revised paper without tracked changes. You should upload this as a separate file labeled 'Manuscript'.

We look forward to receiving your revised manuscript.

Kind regards,

Peter Eickholz

Academic Editor

PLOS ONE

Journal Requirements:

Reviewers' comments:

Reviewer's Responses to Questions

**Comments to the Author**

1. Is the manuscript technically sound, and do the data support the conclusions?

Reviewer #1: No

Reviewer #2: Partly

2. Has the statistical analysis been performed appropriately and rigorously? 

Reviewer #1: No

Reviewer #2: I Don't Know

3. Have the authors made all data underlying the findings in their manuscript fully available?

Reviewer #1: No

Reviewer #2: No

4. Is the manuscript presented in an intelligible fashion and written in standard English?

Reviewer #1: Yes

Reviewer #2: Yes

5. Review Comments to the Author

Reviewer #1: The purpose of this research study was to characterize the pathogenic role of Mycoplasma salivarium and bacterial co-detection patterns on different implant augmentation types. The conclusions are unclear.

Major revision:

A more sophisticated statistical analysis that accounts for the correlation among repeated measures within a patient is needed. Perhaps a mixed linear regression model with factors for time, treatment group, and side would be more appropriate.

Minor revisions:

1- Line 149: “Across” treatment groups.

2- State and justify the study’s target sample size with a pre-study statistical power calculation. The power calculation should include: sample size, alpha level (indicating one or two-sided), minimal detectable difference and statistical testing method.

3- The p-value associated with a correlation is a test of the null hypothesis: correlation equal to zero; however, the absolute magnitude of the coefficient indicates the strength of the linear relationship between two variables. In general, the strength or correlation coefficient is the more important statistic to focus on.

Below is a table for interpreting correlation coefficients:

Coefficient (absolute value) Interpretation

0.90 - 1.0 Very Strong

0.70 - 0.89 Strong

0.40 - 0.69 Moderate

0.10 - 0.39 Weak

less than 0.10 Negligible correlation

Reviewer #2: I have some problems with the major purpose of the study. As there is obviously nothing known about the microbial colonization regarding augmentation, I would not set the primary focus on a bacterial species which was only very rarely described.

Please use the actual term “biofilm” (instead of plaque).

Although, analysis of microbiota was obviously not the primary aim of the study, please mention the primary outcome and the respective power analysis.

Why did the authors focus on the selected microorganisms? I miss Treponema denticola at least. Please provide the detection level of each PCR.

The data on positive results of all analyzed microorganisms should be presented. In my suggestion, these data (although not being statistically significant different between the groups and over time) might be of interest.

The last sentence in the conclusion of the abstract should be weaken.

Did the authors also collect peri-implant sulcular fluid for analysis of certain biomarkers in addition? If yes, inclusion of these data would increase the merit of that manuscript.

6. PLOS authors have the option to publish the peer review history of their article (what does this mean?). If published, this will include your full peer review and any attached files.

Reviewer #1: No

Reviewer #2: No

---

## [Author Response · Author response to Decision Letter 0]

6 Jan 2022

Reviewer #1: The purpose of this research study was to characterize the pathogenic role of Mycoplasma salivarium and bacterial co-detection patterns on different implant augmentation types. The conclusions are unclear.

Major revision:

A more sophisticated statistical analysis that accounts for the correlation among repeated measures within a patient is needed. Perhaps a mixed linear regression model with factors for time, treatment group, and side would be more appropriate.

> We thank the Reviewer for this important suggestion. Delineating interfering random effects on the levels of the variables suggested by the Reviewer from the fixed effects estimated by the model we had included is indeed an important addition. This refers to the second set of analyses in the report, where we pool samples from different groups, sides and timepoints for overall association analyses. Based on the data presented for the first analyses, which are separated by group, side and timepoint, we proposed that random effects would be negligible. We do, however, agree that a more formal assessment of this proposal is warranted. In response to the Reviewer’s suggestion, we now include estimates from a multilevel mixed-effects model with nested random effects for (from highest to lowest level): sampling (tooth or implant), treatment group and timepoint of sampling. The resulting estimates for the fixed effects were indeed very close to the ones from the previous model, indicating that random effects on the variables mentioned did not interfere with the fixed-effects estimations.

Corresponding changes:

Lines 159 – 164

Lines 244 – 247

Minor revisions:

1- Line 149: “Across” treatment groups.

> This has now been corrected.

2- State and justify the study’s target sample size with a pre-study statistical power calculation. The power calculation should include: sample size, alpha level (indicating one or two-sided), minimal detectable difference and statistical testing method.

> The Reviewer raises an important point. Although the setting of the study as a sub-study within an interventional trial was implied in our manuscript, we did not explain this explicitly and we agree that this might be confusing for the reader. The sample-size calculation was strictly performed for the interventional trial and the underlying assumptions have been detailed in the published report for this trial. The sub-study we report on here was designed as exploratory and made secondary use of the sample recruited for the trial, without a pre-specified statistical analysis plan. Obviously, this design carries risks. However, due to the scarcity of previous data on the subject and the need to adapt to the main trial, this was the only feasible design. Incorporating exploratory sub-studies on unexplored questions into interventional trials is common and good practice. These sub-studies have the aim to clarify hypotheses and lead to subsequent, more focused studies. We believe that the study we report on meets this goal, but we do agree that the exploratory nature of the study needed better explanation.

Corresponding changes:

Lines 149 - 151

3- The p-value associated with a correlation is a test of the null hypothesis: correlation equal to zero; however, the absolute magnitude of the coefficient indicates the strength of the linear relationship between two variables. In general, the strength or correlation coefficient is the more important statistic to focus on.

Below is a table for interpreting correlation coefficients:

Coefficient (absolute value) Interpretation

0.90 - 1.0 Very Strong

0.70 - 0.89 Strong

0.40 - 0.69 Moderate

0.10 - 0.39 Weak

less than 0.10 Negligible correlation

> We thank the Reviewer for providing this frame for reference and adapted the wording in the manuscript accordingly.

Corresponding changes: Throughout the paragraphs from line 223 to line 261.

Reviewer #2: I have some problems with the major purpose of the study. As there is obviously nothing known about the microbial colonization regarding augmentation, I would not set the primary focus on a bacterial species which was only very rarely described.

> We agree with the reviewer that the focused interest in exploring interactions of Mycoplasma salivarium may seem uncommon. The senior investigator of the sub-study, Prof. Henrich, dedicated more than 20 years of her academic career to the investigation of roles of Mycoplasma spp. As potential pathogens. Europe’s largest scientific society for clinical microbiology, the European Society for Clinical Microbiology and Infectious Diseases, acknowledges the importance of this work by having formed a dedicated Mycoplasma Research Group. We provide detailed information on the (admittedly few) previous reports on M. salivarium’s role as a possible oral pathogen in the introduction and discussion sections of the manuscript. Although other potential pathogens may feature more prominently in the scientific literature, we believe that exploring M. salivarium’s role is important.

Please use the actual term “biofilm” (instead of plaque).

> The wording in the manuscript has been changed accordingly.

Corresponding changes: Throughout the manuscript.

Although, analysis of microbiota was obviously not the primary aim of the study, please mention the primary outcome and the respective power analysis.

> In line with our response to Reviewer 1, a power or sample size calculation for the sub-study based on pre-analysis considerations is not possible due to the exploratory design. 

Why did the authors focus on the selected microorganisms? I miss Treponema denticola at least. Please provide the detection level of each PCR.

> Individual results for all microorganisms are now provided as part of the anonymised minimal dataset in the supplementary material.

The panel of microorganisms analysed for this study was the result of a previous study referenced in our report (number 26 in our reference list: Schwarz F, Becker K, Rahn S, Hegewald A, Pfeffer K, Henrich B. Real-time PCR analysis of fungal organisms and bacterial species at peri-implantitis sites. International journal of implant dentistry. 2015;1(1):9). We agree with the Reviewer that the panel is not comprehensive. Selection of microorganisms for the panel was based on the aim to include a set of microorganisms known to represent a spectrum of strengths of association with periodontal disease. However, as little is known about pathogens in peri-implant inflammation, even a complete set of microorganisms with strongly established roles as pathogens or indicators of periodontal inflammation would not necessarily be comprehensive. Any panel study can be correctly criticised in this way, and given the wider accessibility of metagenomic sequencing techniques, we believe that future studies, including our own subsequent work, need to employ bias-free or bias-reduced methods.

The data on positive results of all analyzed microorganisms should be presented. In my suggestion, these data (although not being statistically significant different between the groups and over time) might be of interest.

> We thank the Reviewer for this suggestion, data for all analysed microorganisms are now included in supplementary figure 2.

The last sentence in the conclusion of the abstract should be weaken.

> We agree with the Reviewer that a weaker wording more accurately describes the findings of the study and changed the sentence accordingly.

Did the authors also collect peri-implant sulcular fluid for analysis of certain biomarkers in addition? If yes, inclusion of these data would increase the merit of that manuscript.

> Although we agree that this is an excellent idea, we did not measure biomarkers in sulcular fluid and were not able to retain sufficient fluid volume to add these analyses now.

---

## [Decision Letter · Decision Letter 1]

25 Feb 2022

PONE-D-21-31526R1Simultaneous presence of Mycoplasma salivarium and Tannerella forsythia in the sulcus of lateral autogenous root grafts augmented implants is associated with increased sulcus probing depthPLOS ONE

Dear Dr. Kohns Vasconcelos,

Thank you for submitting your manuscript to PLOS ONE. After careful consideration, we feel that it has merit but does not fully meet PLOS ONE’s publication criteria as it currently stands. Therefore, we invite you to submit a revised version of the manuscript that addresses the points raised during the review process. Please ensure that your decision is justified on PLOS ONE’s publication criteria and not, for example, on novelty or perceived impact.

We look forward to receiving your revised manuscript.

Kind regards,

Peter Eickholz

Academic Editor

PLOS ONE

Journal Requirements:

Reviewers' comments:

Reviewer's Responses to Questions

**Comments to the Author**

1. If the authors have adequately addressed your comments raised in a previous round of review and you feel that this manuscript is now acceptable for publication, you may indicate that here to bypass the “Comments to the Author” section, enter your conflict of interest statement in the “Confidential to Editor” section, and submit your "Accept" recommendation.

Reviewer #1: (No Response)

Reviewer #2: (No Response)

Reviewer #3: (No Response)

2. Is the manuscript technically sound, and do the data support the conclusions?

Reviewer #1: Yes

Reviewer #2: Partly

Reviewer #3: Yes

3. Has the statistical analysis been performed appropriately and rigorously? 

Reviewer #1: Yes

Reviewer #2: (No Response)

Reviewer #3: Yes

4. Have the authors made all data underlying the findings in their manuscript fully available?

Reviewer #1: No

Reviewer #2: Yes

Reviewer #3: (No Response)

5. Is the manuscript presented in an intelligible fashion and written in standard English?

Reviewer #1: Yes

Reviewer #2: Yes

Reviewer #3: Yes

6. Review Comments to the Author

Reviewer #1: Minor Revision:

Indicate the underlying covariance structure used in the mixed effects model and the criteria for selecting it.

Reviewer #2: The authors responded to my raised concerns. Answers to minor comments I accept. But still I do not understand the purpose of the study. A bacterium probably of importance in development of oral diseases was investigated related to three augmentation procedures. What is the rationale behind?

Studying the presence and quantity of Mycoplasma salivarium related to disease categories (healthy, mucositis, peri-implantitis, gingivitis, periodontitis) might be of interest. And still, including 36 patients in a clinical research project, would need a primary outcome.

Reviewer #3: This is the long term follow up of a RCT focusing on the efficacy of two different autologous tooth preparations as test group and bone blocks as control (Ref #17). This should be more clearly pointed out in the methods. The present study is no new study, but an important follow up.

As this is a sub group analysis with a descriptional pilot character, the importance of a power calculation from my point of view should not be overestimated.

The title is difficult to understand. I would suggest: “Simultaneous presence of Mycoplasma salivarium and Tannerella forsythia in the sulcus implants after lateral augmentation with autogenous root grafts is associated with increased sulcus probing depth.

Page 2, line 27 & 32: „gingiva“ around implants the term “mucosa” would be more appropriate in the field of implantology.

Materials and Methods are clear and complete.

the results are presented in an appropriate manner.

The discussion addresses all important aspects and covers also the weakness of the study.

The conclusions are supported by the data.

7. PLOS authors have the option to publish the peer review history of their article (what does this mean?). If published, this will include your full peer review and any attached files.

Reviewer #1: No

Reviewer #2: No

Reviewer #3: No

---

## [Author Response · Author response to Decision Letter 1]

11 Apr 2022

We thank the Reviewers for their insightful comments and suggestions, and their continuing support in improving the clarity of the manuscript.

Please find our point-by-point reply to the comments below. Line numbers refer to numbering in the tracked changes version of the revised manuscript.

Reviewer #1: Minor Revision:

Indicate the underlying covariance structure used in the mixed effects model and the criteria for selecting it.

Because possible random effects are nested (i.e. on different levels) but also clearly independent (in that they cannot be associated with one another because the study design defines that both tooth and implant side are sampled at two timepoints in each treatment group, i.e. across these variables data is evenly distributed), selection of different covariance structures did not impact the fixed or random effect estimates. For additional clarity, an independent covariance structure has been specified in the manuscript now (l 168). Yet, using unstructured covariance creates the same results in the model.

Reviewer #2: The authors responded to my raised concerns. Answers to minor comments I accept. But still I do not understand the purpose of the study. A bacterium probably of importance in development of oral diseases was investigated related to three augmentation procedures. What is the rationale behind?

Studying the presence and quantity of Mycoplasma salivarium related to disease categories (healthy, mucositis, peri-implantitis, gingivitis, periodontitis) might be of interest. And still, including 36 patients in a clinical research project, would need a primary outcome.

The primary outcome of the main trial that participants were included into is stated in reference number 17 of the current manuscript (“clinical width of the alveolar ridge”, operationalised as: “The primary endpoint was defined as the clinical width (CW) of the alveolar ridge being sufficient for the placement of an adequately dimensioned dental implant without the need for a secondary grafting at 26 weeks after surgery in either group.”). The nested study that we report on in this manuscript did not independently include patients. Rather, the patient population included in the main trial contributed samples used for this exploratory study. We believe that it is important to explain the intervention used in the main trial in some detail for this manuscript, as potential effects of this intervention are studied here. However, we decided not to elaborate on other aspects of the design of the main trial because they are of little consequence to this study and a reference for the main trial’s report is provided so that interested readers may find more detail there.

In our opinion, investigating abundance of Mycoplasma salivarium in different augmentation procedures makes best use of the background of how the respective samples for this study were obtained. We believe this approach links a broader question of potential roles of M. salivarium in peri-implant disease to a more specific clinical application by demonstrating that abundance of M. salivarium may not simply be something to be observed, but may be dependent on patient management, in this case the use of different augmentation methods. We aimed to interpret the findings with appropriate caution, given that treatment allocation was not randomised, yet there is an association between augmentation method and detection of M. salivarium, which we argue warrants further study.

Reviewer #3: This is the long term follow up of a RCT focusing on the efficacy of two different autologous tooth preparations as test group and bone blocks as control (Ref #17). This should be more clearly pointed out in the methods. The present study is no new study, but an important follow up.

We thank the Reviewer for this suggestion. We added the requested clarification at appropriate points in the introduction and methods sections (ll 72-73, 81 and 94-95). Further, we now stringently refer to the interventional (main) study as “trial” which should also help to delineate mentioning of the trial and the sub-study reported on here.

As this is a sub group analysis with a descriptional pilot character, the importance of a power calculation from my point of view should not be overestimated.

The title is difficult to understand. I would suggest: “Simultaneous presence of Mycoplasma salivarium and Tannerella forsythia in the sulcus implants after lateral augmentation with autogenous root grafts is associated with increased sulcus probing depth.

We adapted the title as suggested with a minor modification (“implant sulcus” instead of “sulcus implants”).

Page 2, line 27 & 32: „gingiva“ around implants the term “mucosa” would be more appropriate in the field of implantology.

We changed the wording in the abstract accordingly.

---

## [Decision Letter · Decision Letter 2]

22 Jun 2022

Simultaneous presence of Mycoplasma salivarium and Tannerella forsythia in the implant sulcus after lateral augmentation with autogenous root grafts is associated with increased sulcus probing depth

PONE-D-21-31526R2

Dear Dr. Kohns Vasconcelos,

We’re pleased to inform you that your manuscript has been judged scientifically suitable for publication and will be formally accepted for publication once it meets all outstanding technical requirements.

Kind regards,

Peter Eickholz

Academic Editor

PLOS ONE

Additional Editor Comments (optional):

Reviewers' comments:

Reviewer's Responses to Questions

**Comments to the Author**

1. If the authors have adequately addressed your comments raised in a previous round of review and you feel that this manuscript is now acceptable for publication, you may indicate that here to bypass the “Comments to the Author” section, enter your conflict of interest statement in the “Confidential to Editor” section, and submit your "Accept" recommendation.

Reviewer #1: All comments have been addressed

Reviewer #3: All comments have been addressed

2. Is the manuscript technically sound, and do the data support the conclusions?

Reviewer #1: (No Response)

Reviewer #3: Yes

3. Has the statistical analysis been performed appropriately and rigorously? 

Reviewer #1: (No Response)

Reviewer #3: Yes

4. Have the authors made all data underlying the findings in their manuscript fully available?

Reviewer #1: (No Response)

Reviewer #3: Yes

5. Is the manuscript presented in an intelligible fashion and written in standard English?

Reviewer #1: (No Response)

Reviewer #3: Yes

6. Review Comments to the Author

Reviewer #1: (No Response)

Reviewer #3: The study objective is clear to me. It is original and interesting. All my concerns have been addressed.

7. PLOS authors have the option to publish the peer review history of their article (what does this mean?). If published, this will include your full peer review and any attached files.

Reviewer #1: No

Reviewer #3: **Yes: **Prof. Al-Nawas

---

## [Editor Report · Acceptance letter]

29 Jun 2022

PONE-D-21-31526R2 

Simultaneous presence of *Mycoplasma salivarium* and *Tannerella forsythia* in the implant sulcus after lateral augmentation with autogenous root grafts is associated with increased sulcus probing depth 

Dear Dr. Kohns Vasconcelos:

I'm pleased to inform you that your manuscript has been deemed suitable for publication in PLOS ONE. Congratulations! Your manuscript is now with our production department. 

Kind regards, 

on behalf of

Dr. Peter Eickholz 

Academic Editor

PLOS ONE